# Interface Design of Head-Worn Display Application on Condition Monitoring in Aviation

**DOI:** 10.3390/s23020736

**Published:** 2023-01-09

**Authors:** Xiaoyan Zhang, Jia’ao Cheng, Hongjun Xue, Siyu Chen

**Affiliations:** 1School of Marine Science and Technology, Northwestern Polytechnical University, Xi’an 710072, China; 2School of Aeronautics, Northwestern Polytechnical University, Xi’an 710072, China

**Keywords:** head-worn display, interface design, condition monitoring, task performance, user preference

## Abstract

Head-worn displays (HWDs) as timely condition monitoring are increasingly used in aviation. However, interface design characteristics that mainly affect HWD use have not been fully investigated. The aim of this study was to examine the effects of several important interface design characteristics (i.e., the distance between calibration lines and the layouts of vertical and horizontal scale belts) on task performance and user preference between different conditions of display, i.e., HWD or head-up display (HUD). Thirty participants joined an experiment in which they performed flight tasks. In the experiment, the calibration lines’ distance was set to three different levels (7, 9 and 11 mrad), and the scale belt layouts included horizontal and vertical scale belt layouts. The scale belts were set as follows: the original vertical scale belt width was set as L, and the horizontal scale belt height as H. The three layouts of the vertical calibration scale belt used were 3/4H, H and 3H/2. Three layouts of horizontal calibration scale belts were selected as 3L/4, L and 3L/2. The results indicated that participants did better with the HWD compared to the HUD. Both layouts of vertical and horizontal scale belts yielded significant effects on the users’ task performance and preference. Users showed the best task performance while the vertical calibration scale belts were set as H and horizontal calibration scale belts were set as L, and users generally preferred interface design characteristics that could yield an optimal performance. These findings could facilitate the optimal design of usable head-worn-display technology.

## 1. Introduction

Head-worn displays (HWDs) have increasingly been applied in aviation for flights over the past two decades [1,2]. With HWDs, flight data can be displayed in three-dimensional stereo to declutter the information presented [3]. HWDs have many advantages over traditional displays, such as increasing situation awareness, greatly reducing the pilot’s workload and improving the ease of mobility [4,5]. As performance indicators and the targeting system are integrated in the HWDs, pilots are able to capture, track and launch missiles to targets on the interface, which enables them to gain an advantage in air battle [6,7]. Efficient and reasonable interface presentation provides pilots with reliable information to improve the task performance [8].

Although HWDs have many potential advantages, if poorly designed, they can also have a number of negative consequences, such as eyestrain, headache, nausea, dizziness and/or disorientation [9,10,11,12,13]. For example, a poorly designed HWD is likely to cause vestibular–visual cue conflict, resulting in cybersickness [14]. About 70% of pilots noted that vision occasionally and unintentionally alternated between their left and right eye either during or after flight. In particular, aviators reported difficulty with making necessary attention switches between eyes [15,16]. In particular, many of the negative consequences can be attributed to improper design of the information display interface of HWDs [14]. Thus, the design of the information presented by HWDs should be addressed.

Interface design is one of the most important research directions in the the HWD field [2]. The most important design parameter of the human–computer interaction interface is the layout of interface information and the information-coding method. Rash et al. conducted experimental studies on HWDs, ergonomics and flight information, encoding the display of a non-fixed-wing aircraft [17]. Van Orden et al. conducted experiments on the shape and color of symbols, to determine their influence on the search time [18]. Andre Wickens et al. studied the information identification of different spatial locations of multiple information channels [19]. According to recent research, HWD interface study focuses on the display mode, character form, display spatial resolution, and contrast of the interface [17,18,19]. The information is mainly presented on the display interface in the form of a scale belt, and the design of the scale belt has a direct impact on the information. However, there are few studies on the carrier of a graduated scale belt in HWDs.

Traditional display research studies these fields in detail. Zhu concluded, through the study of the motion relationship between the scale belt and the pointer in the traditional aircraft cockpit display interface, that there was an interaction between the display position of numbers and the increasing direction of the scale belt [20]. He et al. conducted an experimental study on the thickness, length, interval and observation distance of the scale line on the instrument panel [21]. It was pointed out that the calibration interval had an important effect on the task accuracy [22,23]. Xiong carried out an experimental study on the display position of the heading scale belt in the aircraft cockpit, and proposed a better layout position of the scale belt. Xiong also studied the display position of the heading scale belt in the cockpit and optimized the layout position of the scale belt. Guo divided the plane head-up display into speed, altitude, heading information and other scale belts, and conducted experimental research on different layouts between the digital window and scale belts. Changes to the scale interval were also considered during the experiment, and the layout of the scale belts was optimized.

Visual standards for military aviators were historically set in the 1920s with requirements based on the visual systems of aircrafts at that time, and these standards have changed very little despite significant advances in aircraft technology. Today, pilots are required to perform much more visually demanding tasks in more capable aircrafts using HWDs [24,25,26]. These new visually demanding technologies place previously unconsidered stresses on the human visual system.

In the present literature, there are few studies on the relationship between the scale belt position and scale spacing of a flat panel display. How to apply the standard of a flat panel display to a curved display is also less researched, meaning it cannot be directly applied to HWDs’ interface design. At the same time, there is a lack of research on the influence of scale spacing on the length of the scale belt, or on the characteristics of vertical and horizontal distance layouts of the scale belt (as shown in Figure 1).

## 2. Materials and Methods

### 2.1. Experimental Design

To study the influence between HWDs’ scale belt layout and users’ cognitive processing, this study implemented a four-factor (2 × 3 × 3 × 3) within-subject design, with the display condition, distance between calibration lines, and layouts of vertical or horizontal calibration scale belts serving as the independent variables. The distance between the calibration lines was set at three levels: 7, 9 and 11 mrad (as shown in Figure 1), which was defined as the distance between two adjacent calibration lines. The three layouts of vertical calibration scale belts used were 3/4H, H and 3H/2, which meant the distances from the vertical calibration scale belts to the central point of the display interface were 3H/4, H and 3H/2. The three layouts of horizonal calibration scale belts were selected as 3L/4, L and 3L/2, which meant the distances from the horizonal calibration scale belts to the central point of the display interface were 3L/4, L and 3L/2. The calibration scale belts’ areas were quantified as regular quadrilaterals. H represents the height of the area filled by the calibration scale belts and L represents the width of the area (Figure 2). Nine different combined presentation formats (3 × 3) of layouts of vertical and horizontal calibration scale belts are shown in Figure 3. To compare the differences in user preferences between a flat display and HWD display with the same scale belt layout, this study set up two display conditions: a HWD and normal HUD. A set of user task performance (i.e., task completion time and accuracy rate) and user preference measures were used to assess the various conditions. The task completion time referred to the total time a participant spent completing a task. The accuracy rate was calculated as the proportion of responses that were correct for a task. The user preference for each layout was collected by assessment scales as a subject’s perception of the experiment condition after the experiments.

### 2.2. Participants

Thirty postgraduate students in the School of Aeronautics at Northwestern Polytechnical University (20 males and 10 females, mean age: 22.7 ± 1.6 years) were recruited in this study. All participants were right-handed and had normal or corrected-to-normal vision and healthy upper extremity function. All participants had primary knowledge of cockpit interface design and could effectively use equipment to accomplish simulated flight tasks that simulated real pilots’ behaviors and ensured the reliability of the experimental data. They provided written informed consent before their participation. The study was approved by the Institutional Review Board of the university.

### 2.3. Materials

A software prototype was developed with MATLAB 9.5 to present the task scenarios. The software prototype was performed on a Lenovo computer that was equipped with the Windows 8 operating system (22-inch with a resolution of 1920 × 1080 pixels). The HUD was positioned at a 90° angle with the desk surface, and participants wore the HWD on their head. The task scenario system was installed on two kinds of monitors, respectively, to carry out experiments. The experiment interface was designed according to the basic flight display of an airplane (as shown in Figure 4 for an example). The left calibration scale belts in the display interface showed the change in speed, and its normal value was set at 360~460 km/h. The right calibration scale belts indicated the flight height, normally ranging from 7000~8000 m. The top calibration scale belts referred to the heading angle, and its normal value range was set at 110~190 degrees. The bottom calibration scale belts showed growing speed, and the normal value range was set at 10~30.

### 2.4. Procedures

The study was conducted at the university laboratory. After participants provided informed consent, they were asked to fill out a pre-questionnaire asking their demographic information. Then, a research assistant measured the whole arm’s length from their dominant hand. Participants could adjust the chair to accommodate the experimental condition according to their own preference. Following several practice tasks to familiarize themselves with tasks, participants were asked to touch “ENTER” to initiate the main experimental tasks.

In this experiment, the participants were required to carry out the same monitoring task on two different monitors (HWD and HUD). The mission was to monitor the main flight data information and disturbance changes on the display interface when the aircraft was flying flat at an altitude of about 7500 m, at a speed of about 400 km/h in the cruising state. The participants needed to make quick decisions and carry out operations when there were disturbance changes. The scale spacing and width and height of the scale belt on the monitor changed after each monitoring task. During tasks, if the value shown in the four calibration scale belts changed abnormally, participants were required to press a corresponding direction button on the keyboard (i.e., “↓” or “↑”) to reject the disturbance. For example, if the data on the interface were larger than the threshold set, participants were required to press “↓” to make the data return to normal; otherwise, they pressed “↑”. Abnormal value changes were determined as: Δ > 600 for height, Δ > 60 for speed, Δ > 30 for heading angle and Δ > 10 for growing speed. The tasks were executed both in the HWD condition and normal HUD condition. Data on participants’ performance (i.e., task completion time and accuracy) were automatically recorded by the software prototype. User preference information was collected by assessment scales after the experiments.

The primary task was to monitor the flight information displayed on the four calibration scale belts. The participants were asked to respond as quickly and accurately as possible. The participants were asked to attend the HUD condition experiment first, and after a break, they executed the HWD condition experiment. The vertical distance between calibration lines, layouts of vertical calibration scale belts and layouts of horizonal calibration scale belts were randomized in a full factorial design, and the process was tested for both the HWD and HUD conditions. After completing all tasks, user preference scales were administered to elicit participants’ interface layout preference. The whole experiment could be completed within 1 h.

### 2.5. Data Analysis

The Shapiro–Wilk and Mauchly’s sphericity tests were, respectively, performed to examine whether the task completion time and accuracy rate were normally distributed, and the normality of the task completion time was verified (*p* > 0.05). Four-way repeated-measures analysis of variance (ANOVA)—with the independent variables being the distance between calibration lines, layouts of vertical calibration scale belts, layouts of horizonal calibration scale belts, and display conditions—was performed on the task completion time and user preference. The ANOVA results were validated by Mauchly’s sphericity test. The Greenhouse–Geisser-adjusted degree of freedom and *p* value were used if the sphericity assumption was violated. Post hoc LSD tests with Bonferroni adjustment were also performed where necessary. An α level of 0.05 was adopted for significance. The data analysis was carried out with IBM SPSS version 22 (Chicago, IL, USA).

## 3. Results

### 3.1. Performance

#### 3.1.1. Task Completion Time

Table 1 shows the descriptive analysis and analysis of variance of task completion time under each level of independent variables. The effect of the display condition was significant (F (1, 809) = 187.42, *p* < 0.001). In particular, the task completion time was 20.8% higher for HUD than that for HWD. The effect of the layout of the vertical calibration scale belts was also significant (F = 10.348, *p* = 0.001 < 0.01), while the effect of the layout of the horizonal calibration scale belts and distance between calibration lines was not significant. On average, the task completion time decreased by 5.1% as the vertical distance increased from 3/4H to H and increased by 12.5% from H to 3/2H. Table 2 shows that there was no significant interaction effect (all *p* values > 0.05). Figure 5 shows the effect of calibration lines’ distance and layouts of vertical calibration scale belts on task completion time.

#### 3.1.2. Task Accuracy Rate

Table 3 presents the results of the descriptive analysis and analysis of variance of the task accuracy rate under each level of independent variables. The effect of the display condition was significant (F (1, 809) = 28.823, *p* < 0.001) and the accuracy rate was 5% lower for HUD versus of HWD. The effect of the layout of the horizontal calibration scale belts was significant (F (2, 58) = 2.421, *p* = 0.043 < 0.001), while the distance between the calibration lines and the layouts of the vertical calibration scale belts did not show any significant effect. In particular, when the layout of the horizontal calibration scale belts was L, the accuracy was the highest of the three layouts of the horizontal calibration scale belts. Table 4 shows that there existed significant interaction effects between the distance between the calibration lines and the layouts of the horizonal calibration scale belts; between the distance between the calibration lines and the layouts of the vertical calibration scale belts; and between the layouts of the horizonal calibration scale belts and the layouts of the vertical calibration scale belts. Figure 6 shows the effect of the calibration lines’ distance and the layouts of horizonal calibration scale belts on the task accuracy rate.

### 3.2. User Preferences

As shown in Table 5, all of the interface design factors, i.e., the distance between the calibration lines (F (1.298, 37.649) = 15.606, *p* < 0.001), the layouts of the horizontal calibration scale belts (F (2, 58) = 21.484, *p* < 0.001), and the layouts of the vertical calibration scale belts (F (1.331, 38.6) = 155.477, *p* < 0.001), showed a significant effect on user preferences. On average, the user preference score decreased while the distance between calibration lines increased from 7 mrad to 11 mrad. As for the layouts, users had the greatest preference for L as the layout of the horizontal calibration scale belts, and H derived the highest score for the layout of the vertical calibration scale belts. The reason can be that users favor interface design characteristics that are able to help them achieve better performance and have a better subjective perception. However, there was no significant interaction effect among these three variables (*p* = 0.231 > 0.01).

## 4. Discussion

The rapid development of HWD technology has led to its wide applications in military and aviation settings. HWDs improve search times and potentially improve the overall performance. However, to use it in an advantageous way, the technology should be designed to optimally support the user performance and elicit favorable perceptions. In light of this, the present study was conducted to examine the effects of display conditions and three key interface design characteristics (i.e., distance between calibration lines and layouts of vertical and horizonal calibration scale belts) during flight simulation tasks. Both layouts of vertical and horizontal calibration scale belts were found to interact with the distance between calibration lines. In general, the layouts of vertical and horizontal calibration scale belts yielded significant effects on the performance in flight simulation tasks, while the distance between calibration lines alone had no measurable effect.

### 4.1. Discussion on HUDs and HWDs

We found that for the same information rendered on the interface, the HWD performed better than the HUD. In particular, the task completion time with HWDs was shorter than that with HUDs, and the accuracy was higher. Furthermore, users preferred to use HWDs versus HUDs. With the progress of science and technology, especially the application of off-axis weapons, the shortcomings of HUDs have been exposed. The advantages of HWDs over HUDs are mainly reflected in the field of view and off-axis emission. HWDs overcome the shortcomings of HUDs and allow the pilot to capture information more quickly. When the pilot uses HUDs and a down-view display together, they must constantly rotate their head to observe the environment, which increases their fatigue and the possibility of error during operation. On the other hand, HWDs stop the pilot from losing focus and give the pilot real-time knowledge of the situation.

### 4.2. Discussion of Scale Belt Layout of HWDs

In the experiment, the calibration lines distance was set to three different values (7, 9, and 11 mrad), and the scale belts were set to three different horizontal and vertical intervals, respectively. The horizontal and vertical spacing of the scale belts were set as follows: the original vertical scale belt width was set as L and the horizonal scale belt height as H, then 3/4 and 6/4 of the height and width were selected as the contrast values for the original interface.

In the study of the scale line width, the researchers found that the task completion time was the least when the scale line spacing was 7 mrad. Based on the accuracy data analysis, there was no obvious accuracy difference among the three, but 9 mrad had the highest accuracy. Based on the analysis of the users’ preferences, the 7 mrad and 9 mrad scale lines were significantly preferred over 11 mrad. Overall, the best layout was determined to be the 7 mrad scale line. This is a similar finding to the conclusion of Jiang Shao and colleagues [27]. Their study divided the scale line into four groups of different spacing experiments, which led the researchers to find that the line width had a significant impact on the accuracy and response. The paper indicated that the line width of 6 mrad was best, which is close to the conclusion of 7 mrad obtained in this paper. Shao’s research also showed that the line-width level of the target symbol was extremely important when designing such an augmented reality interface. A reasonable line width could effectively reduce pilots’ cognitive load and improve the efficiency of the interface of avionics system.

In this study of the effect of the layout and spacing of scale belts based on reaction time, we found that when the vertical interval was 3H/4 or H, the reaction time was significantly slower than when it was 3H/2. Based on the analysis of accuracy data, the difference between the horizontal and vertical spacing was not obvious. Based on user preferences, the subjective perception score was significantly higher with the horizontal spacing as L or 3L/2, rather than with 3L/4. Thus, when the horizontal spacing is between L and 3L/2, people have a better subjective perception. The scores for the vertical spacing at 3H/4 or H were significantly higher than those for 3H/2, meaning that people perceive the vertical spacing of 3H/4 or H to give better visual perception. Above all, this study found that the vertical distance has a greater influence on the task completion time than the horizontal distance. Users performed better with a larger horizontal scale belt distance and narrower vertical scale belt. This finding is similar to the results of Wu and colleagues [28]. In Wu’s experiment, the spacing was divided into 12mm, 48 and 96mm, and the length of the main display area set as 39 mm. The results indicated that while the spacing was set at 48mm, the reaction time was shortest and the accuracy was highest among the three sets. Besides this, in Zhu’s study, with an increase in the spacing, the reaction time decreased and the error rate decreased [29].

In conclusion, when the task completion time is taken as the evaluation index, 7 mrad of the scale line width, H of the scale belt width, and L of the scale belt height are better. When the accuracy is taken as the evaluation index, 9 mrad of the scale line width, H of the scale belt width, and L of the scale belt height are better. When users’ preference is taken as the evaluation index, 7 mrad of the scale line width, H of the scale belt width, and L of the scale belt height are better. For the interface distribution coding of a HWD system, 7 mrad and 9 mrad are determined to be the better values for scale line spacing in system application. L and 3L/2 are the best values for the interval of graduated scale belts in the horizontal direction of system application, while 3H/4 and H are the best values of the scale belt spacing in the vertical direction in system application.

Since user preference measures are largely overlooked in the existing literature, this study demonstrated that the majority of participants prefer a distance between calibration lines of 7 mrad, and a medium distance from the calibration scale belts to the central point of the interface (i.e., H for the layout of vertical calibration scale belts and L for the layout of horizontal calibration scale belts). It appears that users favor interface design characteristics that are able to achieve better performance, which is consistent with the findings of previous studies [30,31,32]. New materials can also improve the user preference of wearable electronics such as HWDs. According to recent research, a self-healing multifunctional film can be applicated in HWDs, which can not only expand the device lifetime and reliability, but also bring antibacterial ability [33,34].

## 5. Conclusions

In this paper, through the monitoring of HUD and HWD’s interface in the cruising state, an experiment was conducted on the scale intervals of different scale belts, as well as the horizontal and vertical distances. When carrying out the same task, compared with a HUD display, HWDs had a shorter task completion time and higher accuracy rate, and the user preference was obviously better than that of the HUD display. The optimal layout of the scale belt was obtained by analyzing indexes such as the task completion time and operational accuracy. The scale belt based on this design can meet the needs of users and is consistent with the test results of users’ preferences.

Our study has some limitations. Firstly, the interaction effect of scale spacing and scale belt position variables needs further research. Secondly, more task types simulating different flight periods should be tested for the improvement in HWD interface design. Thirdly, other factors in HWD interface design should be studied to better assist the development of HWD technology. Overall, our study forms a solid basis for extending research into the interface design of HWDs and can contribute to design guidelines for a more effective view of management systems of HWDs. The results can also encourage new studies to be carried out to support the effective and safe adoption of HWD technology.

## Figures and Tables

**Figure 1 sensors-23-00736-f001:**
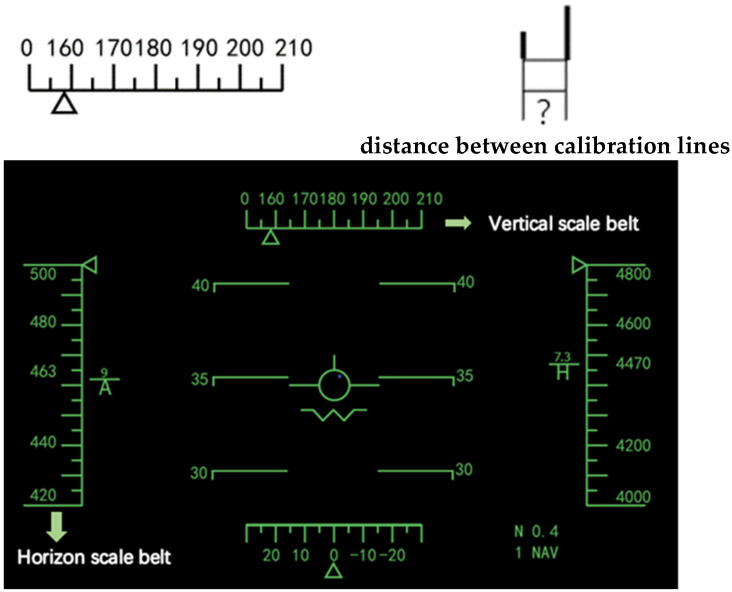
Calibration scale belt (vertical and horizon) and distance between calibration lines.

**Figure 2 sensors-23-00736-f002:**
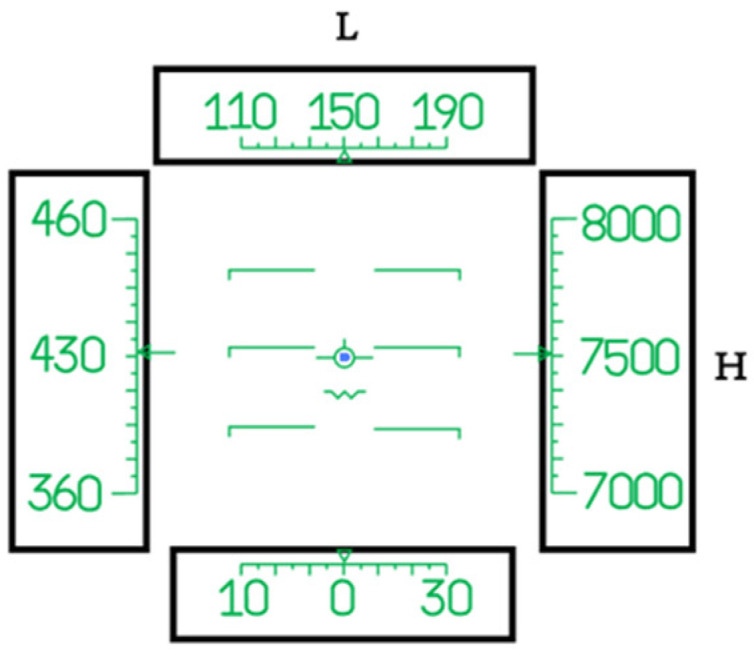
HWD system interface information layout design.

**Figure 3 sensors-23-00736-f003:**
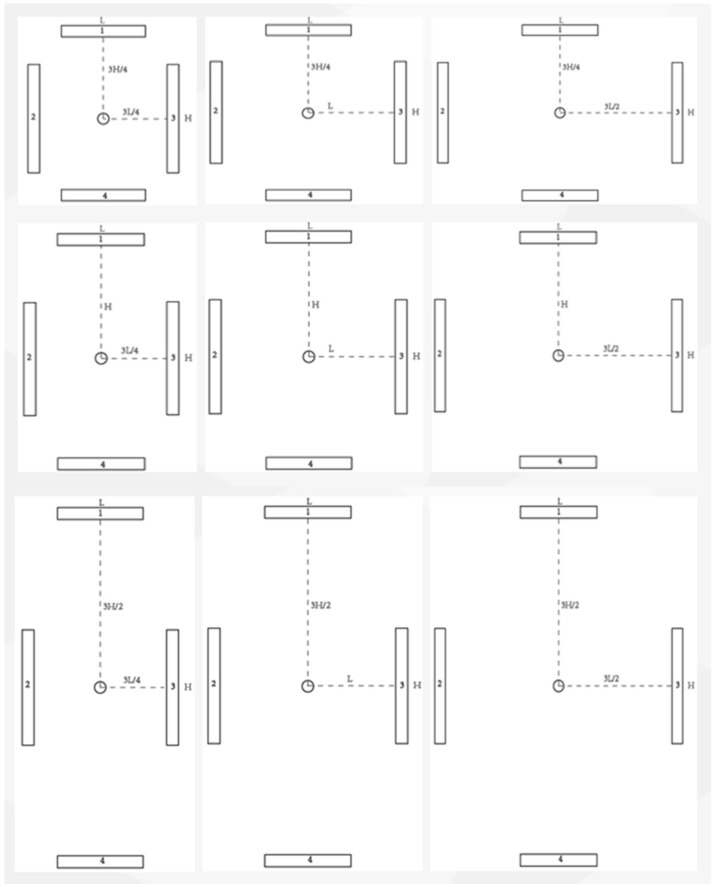
Layouts of vertical calibration scale belts: 3H/4, H, 3H/2 and layouts of horizontal calibration scale belts: 3L/4, L, 3L/2.

**Figure 4 sensors-23-00736-f004:**
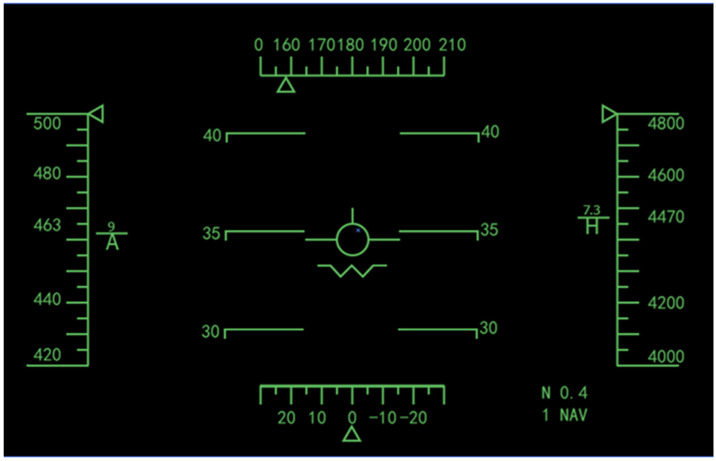
The simulation flight interface.

**Figure 5 sensors-23-00736-f005:**
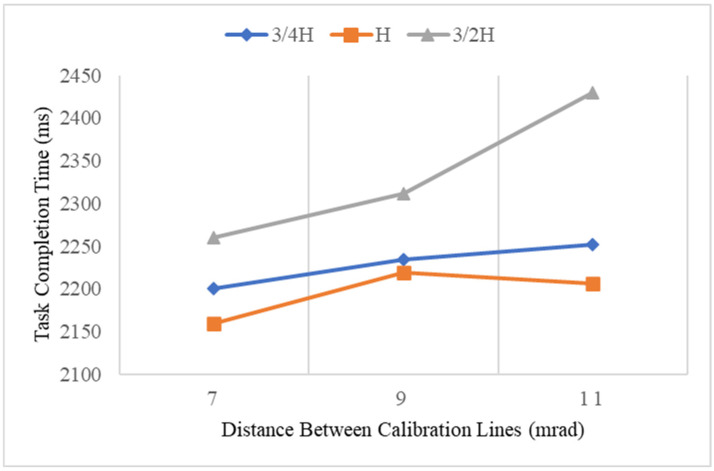
Task completion time (ms) by distance between calibration lines and layouts of vertical calibration scale belts.

**Figure 6 sensors-23-00736-f006:**
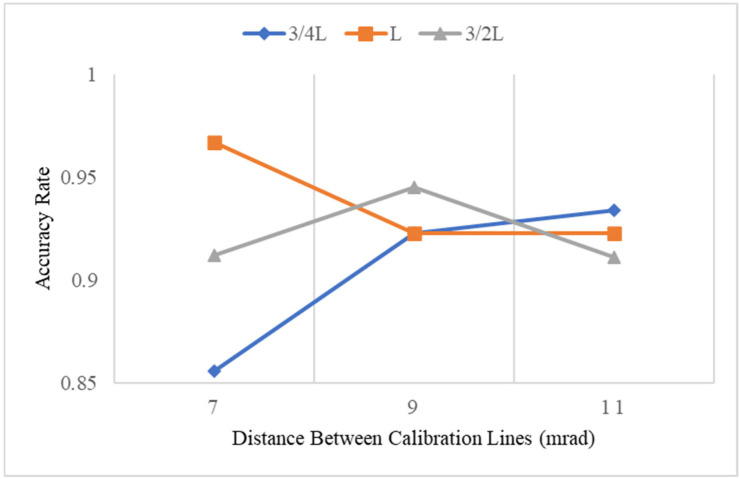
Task accuracy rate by distance between calibration lines and layouts of horizonal calibration scale belts.

**Table 1 sensors-23-00736-t001:** Main effects of display condition, distance between calibration lines, and layouts of vertical and horizonal calibration scale belts on task completion time.

Independent Variables	Task Completion Time (ms)		ANOVA	
	Descriptive Analysis			
	Mean	SE	F Value	*p*-Value
Display condition			187.420	*p* < 0.001
HWD	2241.65	22.64		
HUD	2707.79	24.53		
Distance between calibration lines (mrad)			0.239	0.790
7	2201.61	106.17		
9	2244.07	113.78		
11	2279.26	112.81		
Layouts of vertical calibration scale belts			10.834	0.001
3H/4	2225.90	116.00		
H	2117.22	98.33		
3H/2	2381.82	118.43		
Layouts of horizonal calibration scale belts			2.270	0.080
3L/4	2228.38	107.66		
L	2217.61	115.51		
3L/2	2278.95	109.59		

**Table 2 sensors-23-00736-t002:** ANOVA for task completion time.

Source	F	*p*-Value
Distance between calibration lines	0.239	0.790
Layouts of vertical calibration scale belts	10.834	<0.001
Layouts of horizonal calibration scale belts	0.633	0.543
Distance between calibration lines × Layouts of horizonal calibration scale belts	2.27	0.080
Distance between calibration lines × Layouts of vertical calibration scale belts	0.258	0.903
Layouts of horizonal calibration scale belts × Layouts of vertical calibration scale belts	0.458	0.766
Distance between calibration lines × Layouts of horizonal calibration scale belts × Layouts of vertical calibration scale belts	1.028	0.423

**Table 3 sensors-23-00736-t003:** Main effects of display condition, distance between calibration lines and layouts of vertical and horizonal calibration scale belts on task accuracy rate.

Independent Variables	Task Accuracy Rate		ANOVA	
	Descriptive Analysis			
	Mean	SE	F Value	*p*-Value
Display condition			28.823	<0.001
HWD	0.922	0.006		
HUD	0.876	0.007		
Distance between calibration lines (mrad)			1.168	0.318
7	0.912	0.029		
9	0.930	0.024		
11	0.923	0.029		
Layouts of vertical calibration scale belts			2.421	0.098
3H/4	0.901	0.031		
H	0.938	0.023		
3H/2	0.926	0.028		
Layouts of horizonal calibration scale belts			3.317	<0.001
3L/4	0.904	0.032		
L	0.938	0.023		
3L/2	0.923	0.027		

**Table 4 sensors-23-00736-t004:** ANOVA for task accuracy rate.

Source	F	*p*-Value
Distance between calibration lines	4.851	0.021
Layouts of vertical calibration scale belts	48.236	<0.001
Layouts of horizonal calibration scale belts	6.659	<0.001
Distance between calibration lines × Layouts of horizonal calibration scale belts	3.430	0.018
Distance between calibration lines × Layouts of vertical calibration scale belts	8.857	<0.001
Layouts of horizonal calibration scale belts × Layouts of vertical calibration scale belts	3.360	0.020
Distance between calibration lines × Layouts of horizonal calibration scale belts × Layouts of vertical calibration scale belts	1.356	0.231

**Table 5 sensors-23-00736-t005:** Main effects of distance between calibration lines and layouts of vertical and horizonal calibration scale belts on user preference.

Independent Variables	User Preference		ANOVA	
	Descriptive Analysis			
	Mean	(SE)	F Value	*p*-Value
Distance between calibration lines (mrad)			15.606	<0.001
7	5.93	0.247		
9	5.76	0.212		
11	5.16	0.259		
Layouts of vertical calibration scale belts			21.484	<0.001
3H/4	6.17	0.233		
H	6.64	0.230		
3H/2	4.07	0.254		
Layouts of horizonal calibration scale belts			155.477	<0.001
3L/4	5.20	0.225		
L	5.94	0.245		
3L/2	5.71	0.247		

## Data Availability

Not applicable.

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
