# Peer review of "Interface Design of Head-Worn Display Application on Condition Monitoring in Aviation"

_sensors, 2023, doi:10.3390/s23020736_

Round 1

Reviewer 1 Report

1.     Different citation styles were used in different places. Please make the citation style consistent through the entire manuscript.

2.     According to the recent research, HWDs’ interface study focuses on the display mode, character form, display spatial resolution and contrast of the interface. Citations are needed.

3.     In some places, authors used distance between calibration lines, while in other places, authors used distance between scale marks. Are they the same thing? Also, authors are recommended to show an example of this variable. Otherwise, it is difficult for audience to follow.

4.     2.2 Participants: “The study was conducted at the university laboratory. After participants …” This information should be presented in the 2.3 Procedures.

5.     2.2 Participants: How did the authors determine the sample size of the study?

6.     2.3 Some information presented in 2.3 Procedures should be presented in Materials section. Authors are recommended to have a Material/Stimuli section to present materials used in the study.

7.     Why all participants were asked to attend the HUD condition first and then HWD condition? Why did not the authors balance the sequence of the two conditions?

8.     Results: for some results, the authors kept three digits after the decimal points, but for the others, the authors kept two decimals.

9.     Table 1, there is a star after the p-value of 0.001, what does it mean? Why there were no stars for other p values that smaller than 0.05?

10.   Some of the results presented in Table 1 have been repeatedly presented in Table 2. (same for Tables 3 and 4)

11.   For those significant interactions, did the authors further analyze the simple effect?

12.   Method: How did the authors measure user preferences?

13.   What is the theoretical contribution of the study?

14.   What are the limitations of the study?

Author Response

The authors appreciate the constructive comments and suggestions made by the reviewer. We have tried to address and made response to each of them and trust the quality of the manuscript can be enhanced. Please see the attachment.

Reviewer 2 Report

This manuscript reports effects of several interface design characteristics (i.e., the distance between scale marks, layouts of vertical and horizontal scale belts) on task performance and user preference between Head-worn displays (HWD) and head-up display (HUD). The results show that HWDs have a shorter task completion time and higher accuracy rate than that of the HUD display. Also, the user preference is better. Furthermore, the optimal layout of the scale belt was obtained by analyzing indexes such as the task completion time and operational accuracy. I would recommend it for publication in Sensors after revision as follows:

(1) What is the basis for setting the scale belts? Whether the test sequence of HUD and HWD affects the results should be taken into account.

(2) In section 3.1, the chart comparison is not obvious, can the author change the form of data visualization?

(3) In section 3.2, can you explain why this layout is preferred by users rather than simply describing the results, such as from the perspective of human visual system ?

(4) There are a few grammatical errors in the manuscript, such as 4.1. Discussion on HUD and HWDs.

(5) Some related references need to be cited, such as Adv. Funct. Mater. 2021, 31: 2011133; ACS Appl. Mater. Interfaces. 2022, 14(18): 21509-20.

Author Response

The authors appreciate the constructive comments and suggestions made by the reviewer. The authors have tried to address and made response to each of them and trust the quality of the manuscript can be enhanced. Please see the attachment.

Reviewer 3 Report

This study examined the effects of several important interface design characteristics (i.e., the distance between scale marks, layouts of vertical and horizontal scale belts) on task performance and user preference between different conditions of display. The topic is interesting and the manuscript is well organized.

Figure 5 can be improved. 

The limitation of this study should be discussed in the Conclusion section. 

Author Response

(The authors gave the same response as above.)

Round 2

Reviewer 2 Report

The manuscript can be acceptable.